# A Community-Based Prostate Cancer Screening and Education Program for Asian American Men in Medically Underserved Communities

**DOI:** 10.3390/ijerph21040415

**Published:** 2024-03-28

**Authors:** Dalnim Cho, Beverly Gor, Hyunsoo Hwang, Xuemei Wang, Mike Hernandez, Lovell A. Jones, Jacqueline Frost, Pamela Roberson, Curtis A. Pettaway

**Affiliations:** 1Department of Health Disparities Research, The University of Texas MD Anderson Cancer Center, Houston, TX 77030, USA; dcho1@mdanderson.org (D.C.); beverlygor50@gmail.com (B.G.);; 2Department of Biostatistics, The University of Texas MD Anderson Cancer Center, Houston, TX 77030, USA; 3Department of Urology, The University of Texas MD Anderson Cancer Center, Houston, TX 77030, USA

**Keywords:** prostate cancer screening, prostate cancer knowledge, Asian American men, ethnicity, community education

## Abstract

This study analyzed data from a community-based prostate cancer (PCa) education and screening program (Prostate Outreach Project; POP) to enhance PCa-related knowledge among medically underserved Asian American men. It also examined PCa screening history, clinical abnormalities based on prostate-specific antigen (PSA) tests and digital rectal examination (DRE) results, and follow-up and PCa diagnosis rates. Participants—521 Asian men (251 Vietnamese, 142 Chinese, and 128 South Asians)—were offered PCa screening using PSA tests and/or DRE and an educational session on PCa. Of these men, 277 completed PCa-related knowledge surveys before and after viewing an educational video. Significant between-group differences in PCa-related knowledge were found at pre-assessment (*p* < 0.001) but not at post-assessment (*p* = 0.11), at which time all groups showed improved PCa-related knowledge. Most participants (77.9%) had never received PCa screening, but Vietnamese men had the lowest previous screening rate (17.3%). Chinese men had elevated PSA values and the highest abnormal DRE rates. Of the 125 men with abnormal screening outcomes, only 15.2% had adequate follow-up. Of the 144 men diagnosed with PCa in POP, 11.1% were Asians (seven Chinese, six Vietnamese, and three South Asian). Despite the ethnic heterogeneity among Asian men, a community outreach program may successfully enhance their PCa-related knowledge.

## 1. Introduction

Prostate cancer is the most commonly diagnosed non-skin cancer among Asian men residing in the United States [1,2]. Prostate cancer incidence is projected to rise among Asian Americans due to its correlation with aging [3]. Notably, the median age at prostate cancer diagnosis is 67 years old [4], and the number of Asians aged 65 and older is rapidly growing in the United States; a 102% increase is expected between 2019 and 2040 [5]. This demographic shift underscores the growing concern for prostate health among Asian American men and the urgent need for targeted health interventions and research. Despite this pressing need, there has been a noticeable shortfall in efforts to enhance prostate health in Asian American men, resulting in an underrepresentation of these men in prostate cancer-related research.

Community-based cancer screening and education programs offer a critical opportunity to improve men’s prostate cancer-related knowledge. These community-based programs address challenges like transportation and time, which are often reported as reasons for Asian Americans’ low participation rates in clinical trials and research [6,7,8]. These programs are also of significant benefit to medically underserved Asian American populations because they reduce barriers to healthcare and information access. However, most existing community-based prostate cancer screening and education programs fail to include a representative number of Asian American men [9,10,11]. Consequently, the effectiveness of these programs in improving prostate cancer-related knowledge in Asian American populations remains largely unknown.

Another significant limitation in prostate cancer-related studies is the lack of disaggregated data for the Asian race and ethnicity. Often, Asian Americans are combined with Pacific Islanders [12,13,14,15], which hinders the ability to understand the specific characteristics and needs of Asians. For instance, national studies have shown lower prostate cancer screening rates among Asian American and Pacific Islander men compared to non-Hispanic Whites [13,14,15]. However, grouping Asians and Pacific Islanders obscures potential screening rate differences between these racial groups and the variations within diverse Asian subgroups. Data disaggregation by Asian ethnicity is crucial to grasp the prostate cancer burden among Asian men, given their diverse origins, languages, English proficiency, education, and income levels [16]. Thus, studies aggregating Asians as a single group may overlook subgroup distinctions, rendering their findings less applicable to the entire Asian American population.

To address these critical knowledge gaps, the authors of the present study conducted a secondary data analysis of the results for three large Asian American ethnic subgroups—Vietnamese, South Asian (Asian Indian/Pakistani), and Chinese men—who were included in a community-based prostate cancer screening and education program called the Prostate Outreach Project (POP) [17]. The POP was provided in medically underserved communities, which were broadly defined to include populations such as the underinsured, uninsured, those with low education or socioeconomic status, and residents of inner-city areas, as well as the unemployed [18].

The primary aim of this study was to evaluate the effectiveness of the POP in improving prostate cancer-related knowledge among Asian men. This aim included an investigation of potential differences between the three ethnic groups. The two secondary aims were to identify the rates of prostate cancer screening and the prevalence of clinical abnormalities through baseline prostate-specific antigen (PSA) and digital rectal examination (DRE) findings, and to determine the rates of follow-up and prostate cancer diagnosis among Asian men who participated in the POP. It was hypothesized that the men’s prostate cancer-related knowledge would be improved after their participation in an educational session. Additionally, we expected lower prostate cancer screening and incidence rates in our sample of Asian men compared to the national data; the prostate cancer screening rate in the United States was reported as 52.1% in 2004 [19], and the prostate cancer incidence rate among Asian American and Pacific Islander men was 2.0% during 2003–2017 [12]. Due to limited prior research, we refrained from firm hypotheses on knowledge change, screening rates, clinical outcomes, follow-up rates, and prostate cancer diagnosis disparities across ethnicities.

## 2. Materials and Methods

### 2.1. Participants and Procedures

Between 2003 and 2008, The University of Texas MD Anderson Cancer Center conducted the POP among the medically underserved in Harris County, TX, USA. Originally aimed at Black men in such communities, the program subsequently expanded to include Hispanic, Asian, and non-Hispanic White men due to perceived necessity. MD Anderson Cancer Center’s Institutional Review Board approved the study, and all participants provided written informed consent.

The design of the POP has been extensively described elsewhere [17]. Briefly, for this portion of the study, a mobile unit facilitated screening and education sessions at various venues where participants were already gathered for planned activities (e.g., churches, community centers, and grocery stores). The POP participants were offered free prostate cancer education and screening. Therefore, the POP effectively addressed barriers to access (e.g., transportation, time, and costs). Prostate cancer education used video content developed by a multidisciplinary panel, including medical oncologists, radiation oncologists, urologists, medical illustrators, and the institution’s public education department. The content covered prostate cancer prevention, early detection risks and benefits, and treatment options. Narration was available in English, Spanish, Chinese, and Vietnamese so that the educational materials would be accessible to the speakers of multiple languages. Additionally, bilingual volunteers and staff aided translation for limited English proficiency individuals.

According to the guidelines used at the time of the study [20], men opting for testing underwent a PSA blood test with or without a DRE. A trained phlebotomist drew 10 mL of blood for PSA analysis, deeming results ≥ 4 as abnormal [21]. A urologist or physician’s assistant conducted the DRE. A DRE was considered abnormal if a prostate nodule or induration was detected upon palpation. Men received screening results along with follow-up options, if necessary, through mailed letters. If the clinical findings were suggestive of disease, participants received counseling over the telephone from POP staff and were given information about resources for receiving quality care and follow-up of test results. Uninsured individuals were referred to the county’s indigent healthcare system through a partnership with Harris Health. Adequate follow-up was defined as the seeking of guidance from a physician regarding abnormal test results. Initiatives aimed to connect those with abnormal results to physician evaluations, spanning at least 6 months before categorizing them as lost to follow-up. Communication involved telephone calls and mailed reminders to participants.

Participants completed a 3-page survey capturing personal demographics (e.g., age, ethnicity, education, and income), care access (e.g., primary care provider and health insurance), and prostate cancer screening history. Their prostate cancer-related knowledge (e.g., ‘A man is more likely to develop prostate cancer if his father had it’) was evaluated using a 10-item survey (yes/no questions) from a previous study [22], which was taken before and immediately after watching the educational video. Correct answers before and after the session were compared to determine the effectiveness of the video as an educational tool. Translations were provided for Spanish, Chinese, and Vietnamese versions of the surveys.

Finally, the prostate cancer endpoint included men diagnosed with prostate cancer, irrespective of diagnosis timing within or after the POP follow-up period. For noncompliant POP participants diagnosed with prostate cancer outside the program, the Texas Cancer Registry—a statewide, population-based registry—was utilized to identify prostate cancer cases from April 2003 to December 2011.

### 2.2. Statistical Analysis Plans

In terms of data analysis, categorical variables were summarized using frequencies and percentages, with chi-squared tests comparing ethnic subgroups. Continuous variables were summarized using means (with Standard Deviations; SDs) or medians (with interquartile ranges; IQRs), and the Kruskal–Wallis test or Wilcoxon rank-sum test was employed for comparisons, as suitable. Changes in prostate cancer-related knowledge before and after the POP program were assessed using the Wilcoxon signed-rank test. Additionally, a multivariable linear model was used to assess the association between ethnicity and pre- and post-education knowledge scores, adjusting for education level. All analyses were conducted in R (version 4.2.1), with statistical significance set at *p* = 0.05.

## 3. Results

### 3.1. Participant Characteristics

Of the 4420 men who received prostate cancer screening over the 5-year study period, 521 Asian American men (11.8%), consisting of 251 Vietnamese, 142 Chinese, and 128 South Asian men, were included in the study. Table 1 presents participant demographics and prostate cancer screening data. The median age of the Chinese men (63 years) was higher than that of the Vietnamese (59 years) and South Asian (54 years) men. Chinese men showed a higher proportion (48.2%) with bachelor’s or advanced degrees compared to South Asian (37.4%) and Vietnamese (10.5%) men. Over half (59.7%) of Vietnamese men reported annual household incomes < $15,000, as compared to 47.7% of Chinese and 21.8% of South Asian men. Although most lacked a primary care provider and health insurance, statistically significant group differences existed in health insurance coverage (*p* < 0.001): Chinese men had the highest rate of health insurance coverage (44.7%).

### 3.2. Improvement in Prostate Cancer-Related Knowledge

Table 2 displays prostate cancer-related knowledge scores before and after watching the educational video, along with associated score changes. A subset of participants (n = 277: 101 Chinese, 156 Vietnamese, and 20 South Asian men) completed both pre- and post-surveys. Combining ethnic subgroups, the median correct answers for pre- and post-assessment were 4 (IQR, 2–7) and 7 (IQR, 6–8), respectively (*p* < 0.001). The median pre-assessment correct answers were 4 (IQR, 2–6), 5 (IQR, 2–7), and 1 (IQR, 0–3) for Chinese, Vietnamese, and South Asian men, respectively. There were statistically significant differences in pre-education knowledge levels among ethnic groups (*p* < 0.001). For post-assessment, the median correct answers were 7 (IQR, 6–8), 8 (IQR, 5.75–9), and 6 (IQR, 3–8) for Chinese, Vietnamese, and South Asian men, respectively, though statistical significance was not observed (*p* = 0.11). Comparable results arose when excluding South Asian men and solely comparing Chinese and Vietnamese men; the resulting *p* values were 0.03 and 0.58, respectively.

Additionally, the multivariable linear model, which included educational level, yielded comparable results (see Table 3). In this model, both ethnic group and education level were significantly associated with pre-education knowledge scores. Specifically, Chinese (*p* = 0.01) and South Asian (*p* < 0.001) individuals, as well as those without a bachelor’s degree (*p* = 0.04), reported lower scores compared to their Vietnamese counterparts. However, the post-education analysis revealed that neither ethnicity (with *p*-values of 0.58 for Chinese and 0.35 for South Asians) nor education level (*p* = 0.62) were significantly associated with post-education knowledge scores.

### 3.3. Prostate Cancer Screening and PSA/DRE Results

Most participants (77.9%) reported no prior prostate cancer screening. The difference between subgroups was statistically significant: Vietnamese men had the highest proportion (82.7%) never screened, followed by South Asian men (75.3%) and Chinese men (71.2%). Table 4 reports findings on abnormal prostate cancer screening results. Chinese men exhibited the highest median PSA values (1.4; range = 0.1–18.2), followed by Vietnamese (0.9; range = 0.1–134) and South Asian men (0.7; range = 0.1–8.5). Similarly, Chinese men displayed the highest proportion of abnormal PSA results (15.5%), followed by South Asian (7.9%) and Vietnamese (6.5%) men. Chinese men exhibited a greater percentage of abnormal DRE outcomes (22.4%) compared to Vietnamese (6.6%) and South Asian (4.8%) men. Furthermore, Chinese men reported a higher percentage of abnormal findings in both PSA and DRE tests (6.7%) than Vietnamese (1.3%) and South Asian (1.2%) men. However, the heightened clinical abnormalities among the Chinese men may be related to their older age (median age, 63 years for Chinese men vs. 59 years for Vietnamese men and 54 years for South Asian men), as age is an established risk factor for elevated PSA levels [23,24].

### 3.4. Follow-Up and Prostate Cancer Incidence

A total of 125 Asian men (56 Chinese, 51 Vietnamese, and 18 South Asian) had abnormal screening results. Merely 19 (15.2%) underwent satisfactory follow-up with a physician (12 Chinese, 4 Vietnamese, and 3 South Asian). However, no significant differences emerged between the three ethnic groups (*p* = 0.16). Of the 144 men diagnosed with prostate cancer in the POP study, 16 (11.1%) were Asians (7 Chinese, 6 Vietnamese, and 3 South Asian). Due to these limited numbers, we refrained from testing between-group differences.

## 4. Discussion

As hypothesized, the POP successfully enhanced prostate cancer-related knowledge among our Asian male participants (*p* < 0.001). Before the program, prostate cancer-related knowledge significantly varied among subgroups and by education level. South Asians exhibited the lowest knowledge (median correct answer = 1), followed by Chinese (median = 4) and Vietnamese men (median = 5). Men with less than a bachelor’s degree reported lower knowledge scores compared to those with at least a bachelor’s degree. However, after the program, knowledge improved across groups and these differences vanished (median correct answers = 7 for Chinese, 6 for South Asians, and 8 for Vietnamese). That is, neither ethnicity nor education level were significantly associated with prostate cancer-related knowledge at post-education. These results suggest that the POP was effective in improving prostate cancer-related knowledge among these 3 distinct Asian ethnic groups and across various education levels.

The success of POP in improving prostate cancer-related knowledge among Asian men holds particular significance because these men were recruited from medically underserved communities, a largely overlooked population in prostate cancer-related research. Notably, 72.9% had education below a bachelor’s degree, 47.3% had <$15,000 household income, and 65.3% lacked health insurance. Hence, our study involving medically underserved Asian men could counter the model minority stereotype that assumes Asians possess high socioeconomic status and therefore do not require specific programs and interventions for their health and well-being. This perception is not only inaccurate but may also jeopardize the health of Asian Americans and propagate health disparities [25].

Only 22.1% of participants had prior prostate cancer screening, notably lower than the United States population (52.1%) [19]. Despite low prostate cancer screening rates among Asian American men, significant ethnic differences were observed. Vietnamese men had the lowest rate (17.3%), followed by South Asian men (24.8%), while Chinese men exhibited a higher yet still-low rate (28.8% compared to the United States rate of 52.1%). This discrepancy, particularly pronounced in Vietnamese men, may stem from the POP’s focus on medically underserved populations and Vietnamese men’s lower socioeconomic status among the three ethnic groups. Notably, higher education, income, and health insurance have been linked to increased PSA screening among Asian Americans [13].

As this research predates the issuance of the 2018 United States Preventive Services Task Force (USPSTF) revised guidelines for prostate cancer screening [26], it is challenging to interpret participant screening rates according to current guidelines. Present USPSTF recommendations suggest men aged 55 to 69 discuss potential PSA screening benefits and harms with clinicians (Grade C), while not recommending PSA-based screening for those 70 and older (Grade D) [26]. During the time of the POP program, however, the American Cancer Society recommended annual PSA and DRE screenings for men aged 50 or older, and noted that high-risk individuals might require screening starting from age 40 [20]. However, studies have reported that following the implementation of the 2012 USPSTF guidelines, there was an increase in adverse, pathologic prostate cancer found on biopsy [27] as well as an uptick in the incidences of advanced and distant metastatic disease [28]. Because some Asian American subgroups (e.g., Chinese, Indian, and Pakistani men) exhibit advanced prostate cancer at diagnosis more than non-Hispanic Whites [29,30], healthcare providers need to be aware of the overall low rates of prostate cancer screening among Asian American men and subgroup differences in prostate cancer screening.

Regarding follow-up, among those with abnormal results (*n* = 125), only 19 (15.2%) had adequate follow-up. This low follow-up rate might relate to participants coming from medically underserved communities; research indicates ethnic minorities, uninsured/underinsured individuals, or those with lower education often experience inadequate follow-up [31,32,33]. However, these factors cannot fully explain the low follow-up rates among Asian men because, in the POP study, 46.7% of the non-Asian men with abnormal results had adequate follow-up despite being from medically underserved communities. We reason that cultural relevance might be another factor, with the POP potentially being less resonant for Asian men than other ethnicities or races. Although POP staff effectively contacted participants about abnormal results, the study had minimal Asian clinicians for DREs and few Asian American male staff for recruitment and follow-up. This absence of ethnic resemblance between participants and staff could contribute to the low follow-up among Asian American men, as ethnic alignment is pivotal in culturally relevant intervention design [34].

Among the 144 prostate cancer cases identified in the POP study, 16 (11.1%) were among Asians, surpassing the national figure of 2.0% for Asians and Pacific Islanders diagnosed during 2003–2017 [12]. Importantly, our sample covered only three Asian ethnic groups, while national data encompassed all Asian groups and combined Asians with Pacific Islanders. Nonetheless, this high incidence rate warrants cautious interpretation, given non-Hispanic White participants constituted just 7.2% of the POP. Notably, during 2003–2017, non-Hispanic Whites constituted 74% of all prostate cancer diagnoses [12]. Hence, our study’s denominator might be smaller.

The present study has several limitations. Group imbalance existed, with fewer South Asians in the sample, potentially influencing results. Additionally, only 15.6% of South Asian men completed the pre- and post-prostate cancer–related knowledge surveys, possibly due to lacking cultural adaptation of the POP for this group. Unlike the provision of education in Chinese and Vietnamese, South Asian languages were not included. However, no unified language exists among all South Asians, making it impractical to offer the program in numerous languages (e.g., Hindi, Urdu, Punjabi). Additionally, given that this study was conducted prior to the implementation of the current USPSTF prostate cancer screening guidelines—which advise against routine PSA screening due to the risk of considerable overdiagnosis [35]—the prostate cancer screening rates observed may not reflect the current rates among Asian American men. Furthermore, given that Vietnamese is the largest Asian ethnic group in Houston, our sample was mostly Vietnamese; thereby, the population does not align with the United States national proportions of Asian ethnic groups. Finally, the assessment of post-education prostate cancer-related knowledge was conducted immediately after participants viewed the educational video. The immediate assessment might not fully reflect the participants’ long-term knowledge retention, which could partly explain the low follow-up rates observed in this study. However, this methodological decision was driven by the practical constraints of conducting the study within community settings, where it was not feasible to follow up with participants who were recruited on-site for subsequent assessments at multiple times.

Despite these limitations, our study underscores the significance of the POP as one of the rare community-based prostate cancer screening and education initiatives that effectively engaged medically underserved Asian men and provided disaggregated data for three distinct Asian groups. Remarkably, 11.8% of POP participants were Asians, surpassing their representation in the United States population during the POP’s time (4.2% from the 2000 census [36] and 5.6% from the 2010 census [37]). This inclusion remains noteworthy even in today’s context, where the Asian population has grown to 6.0% as per the 2020 census [38]. Additionally, this study is among the few that encompass three distinct Asian ethnic groups and, to the best of our knowledge, it is the first to offer explicit insights into their similarities and differences regarding prostate cancer-related knowledge, clinical abnormalities, prostate cancer screening and follow-up rates, and prostate cancer diagnosis. Moreover, our findings emphasize the varied prostate health profiles within Asian American subgroups, dispelling the notion of a uniform low risk for the disease among all Asian American men. Ultimately, our study underscores the potential of community outreach programs like POP to enhance the prostate health knowledge of diverse Asian ethnic groups from medically underserved communities.

## 5. Conclusions

Among these three Asian ethnic groups, we observed similarities in their overall low rates of prostate cancer screening and follow-up on abnormal test results. Despite the ethnic heterogeneity within these groups, a community outreach program such as the POP can effectively enhance their prostate cancer-related knowledge, which, in turn, may likely benefit their prostate health. We anticipate our study’s outcomes will amplify attention given to prostate health across different Asian American subgroups and encourage heightened efforts to meet the health needs of Asian American men, who remain significantly under-represented in prostate health research. In the future, when designing community-based programs, it is important to proactively devise innovative strategies to enhance follow-up procedures after abnormal prostate cancer screening findings among ethnically diverse Asian individuals. Supporting this, there is an urgent need for research focused on the unique cultural characteristics of Asian American men. Identifying these distinctions compared to other racial/ethnic groups could help explain their cancer screening and other health-seeking behaviors. This, in turn, will support the development and implementation of culturally relevant cancer screening interventions for Asian American men.

## Figures and Tables

**Table 1 ijerph-21-00415-t001:** Demographic characteristics and prostate cancer screening histories of study participants.

	Ethnic Group			*p*-Value	Total *N* = 521
	Chinese *n* = 142	Vietnamese *n* = 251	South Asian *n* = 128		
Median age (min, max)	63 (40, 82)	59 (40, 78)	54 (35, 74)	<0.001 ^a^	59 (35, 82)
Education, *n* (%)				<0.001 ^b^	
<High school	21 (15.8)	84 (34.1)	33 (26.9)		138 (27.5)
High school/GED	13 (9.8)	80 (32.5)	29 (23.6)		122 (24.3)
Some college	35 (26.3)	56 (22.8)	15 (12.2)		106 (21.1)
Bachelor’s degree	36 (27.1)	23 (9.3)	36 (29.3)		95 (18.9)
Advanced degree	28 (21.1)	3 (1.2)	10 (8.1)		41 (8.2)
Missing	9	5	5		19
Household income, *n* (%)				<0.001 ^b^	
<$15,000	51 (47.7)	123 (59.7)	22 (21.8)		196 (47.3)
$15,000–$25,999	20 (18.7)	41 (19.9)	27 (26.7)		88 (21.3)
$26,000–$35,999	14 (13.1)	16 (7.8)	23 (22.8)		53 (12.8)
$36,000–$45,999	6 (5.6)	17 (8.3)	7 (6.9)		30 (7.2)
$46,000–$55,999	3 (2.8)	4 (1.9)	8 (7.9)		15 (3.6)
$56,000–$65,999	5 (4.7)	1 (0.5)	8 (7.9)		14 (3.4)
≥$66,000	8 (7.5)	4 (1.9)	6 (5.9)		18 (4.3)
Missing	35	45	27		107
Primary care physician, *n* (%)				0.128 ^b^	
No	88 (62.0)	172 (68.5)	75 (58.6)		335 (64.3)
Yes	54 (38.0)	79 (31.5)	53 (41.4)		186 (35.7)
Insurance, *n* (%)				<0.001 ^b^	
No	68 (55.3)	139 (60.4)	96 (86.5)		303 (65.3)
Yes	55 (44.7)	91 (39.6)	15 (13.5)		161 (34.7)
Missing	19	21	17		57
Previous prostate cancer screening, *n* (%)				<0.001 ^b^	
No	89 (71.2)	191 (82.7)	73 (75.3)		353 (77.9)
Yes	36 (28.8)	40 (17.3)	24 (24.8)		100 (22.1)
Missing	17	20	31		68

Column percentages may not total 100% due to round-off errors. ^a^
*p*-value from a Kruskal–Wallis test. ^b^
*p*-value from a chi-square test. Abbreviations: GED, General Educational Development.

**Table 2 ijerph-21-00415-t002:** Knowledge about prostate cancer on pre- and post-education tests across ethnic groups.

	No. of Correct Answers, Median (Range)				
Knowledge	By Ethnic Group			Group Difference (*p*-Value ^a^)	All Ethnic Groups, Median (Range)
	Chinese *n* = 101	Vietnamese *n* = 156	South Asian *n* = 20		
At pre-education	4 (2, 6)	5 (2, 7)	1 (0, 3)	<0.001	4 (2, 7)
At post-education	7 (6, 8)	8 (5.75, 9)	6 (3, 8)	0.11	7 (6, 8)

^a^ *p*-value from a Kruskal–Wallis test.

**Table 3 ijerph-21-00415-t003:** Multivariable linear model assessing the association between ethnic group, education level, and pre- and post-education knowledge scores.

	Knowledge at Pre-Education		Knowledge at Post-Education	
Ethnic Group ^a^	Coefficient	*p*-Value	Coefficient	*p*-Value
Chinese	−1.17	0.01	0.24	0.58
South Asian	−2.98	<0.001	−0.72	0.35
Education ^b^	1.01	0.04	0.23	0.62

^a^ reference group = Vietnamese. ^b^ 0 = below a bachelor’s degree; 1 = at least a bachelor’s degree.

**Table 4 ijerph-21-00415-t004:** Clinical characteristics of study participants.

	Ethnic Group			*p*-Value	Total *N* = 521
	Chinese *n* = 142	Vietnamese *n* = 251	South Asian *n* = 128		
Median PSA ^ϯ^ (min, max)	1.4 (0.1, 18.2)	0.9 (0.1, 134)	0.7 (0.1, 8.5)	<0.001 ^a^	1 (0.1, 134)
Abnormal PSA, *n* (%)					
No (<4)	109 (84.5)	231 (93.5)	116 (92.1)	0.014 ^b^	456 (90.8)
Yes (≥4)	20 (15.5)	16 (6.5)	10 (7.9)		46 (9.2)
Missing	13	4	2		19
DRE texture, *n* (%)					
Normal	90 (77.6)	226 (93.4)	80 (95.2)	<0.001 ^b^	396 (89.6)
Abnormal	26 (22.4)	16 (6.6)	4 (4.8)		46 (10.4)
Missing	26	9	44		79
Combined results of DRE and PSA, *n* (%)					
Both normal	73 (69.5)	211 (88.7)	73 (89.0)	<0.001 ^b^	357 (84.0)
Only PSA abnormal	9 (8.6)	13 (5.5)	5 (6.1)		27 (6.4)
Only DRE abnormal	16 (15.2)	11 (4.6)	3 (3.7)		30 (7.1)
Both abnormal	7 (6.7)	3 (1.3)	1 (1.2)		11 (2.6)
Missing	37	13	46		96

Column percentages may not total 100% due to round-off errors. ^ϯ^
*n* = 13, *n* = 4, and *n* = 2 Chinese, Vietnamese, and south Asian men, respectively, did not complete PSA. ^a^
*p*-value from a Kruskal–Wallis test. ^b^
*p*-value from a chi-square test. Abbreviations: DRE, digital rectal exam; PSA, prostate-specific antigen.

## Data Availability

The data presented in this study are available on request from the corresponding author.

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
