# Peer review of "A Community-Based Prostate Cancer Screening and Education Program for Asian American Men in Medically Underserved Communities"

_ijerph, 2024, doi:10.3390/ijerph21040415_

Round 1

Reviewer 1 Report

Comments and Suggestions for Authors

I would like to commend the authors for addressing the issues of screening a particular subset of the US population that is often overlooked in studies. Population-based screening programs should address the specifics of individual communities, some of which are marginalized when it comes to education, financial means, access to health care, and insurance status. Similar to some minorities in Europe (e.g., Roma women who screen for cervical cancer), Asian men appear to be less likely to be screened for prostate cancer in the US and to obtain the necessary follow-up, both due to socio -economically, as well as, due to specific ethnic beliefs and behaviors. I would suggest the authors consider the following manuscript that exemplifies the influence that being part of a particular community can influence screening participation and access to healthcare (Simion, L.; Rotaru, V.; Cirimbei, C.; Gales, L.; Stefan, D.-C.; Ionescu, S.-O.; Luca, D.; Doran, H.; Chitoran, E. Inequities in Screening and HPV Vaccination Programs and Their Impact on Cervical Cancer Statistics in Romania Diagnostics 2023 , 13, 2776. https://doi.org/10.3390/diagnostics13172776).

Having said that, I think this manuscript is of sufficient significance to the scientific community, it is well structured, with a clear methodology and discussion. However, I suggest a few changes:

I think the study has some limitations that were acknowledged by the authors (the most important being the older age of the Chinese respondents and the fact that the South Asians did not benefit from educational materials in their own language). The bias introduced by the age difference should be discussed in the results section, not in the discussion (results presented as such can be a bit misleading) – please correct this. The other issue, although discussed, may also introduce severe bias. Yes, it is true that English is an official language in many South Asian countries, but it is also true that it is usually spoken by more educated people. Since the study included people with a low level of education, this may be a consequence. Another limitation, in my opinion, would be that the proportion of Chinese respondents is lower than the proportion of the US Chinese population (the group is not truly representative of all Asian men living in the US).

In my opinion, a paragraph including some specific cultural peculiarities of the Asian population living in the US that may better explain the lack of tracking is warranted and should be added.

One of the conclusions states: "Differences emerged in clinical abnormalities and baseline knowledge of prostate cancer." Please remove this as the first part can be explained by age difference rather than ethnic group and the second part needs to be further explained (Vietnamese group had a higher rate of correct answer although a level of education lower Please add a multivariate analysis of difference by educational attainment to the results section.

Author Response

Reviewer 1

I would like to commend the authors for addressing the issues of screening a particular subset of the US population that is often overlooked in studies. Population-based screening programs should address the specifics of individual communities, some of which are marginalized when it comes to education, financial means, access to health care, and insurance status. Similar to some minorities in Europe (e.g., Roma women who screen for cervical cancer), Asian men appear to be less likely to be screened for prostate cancer in the US and to obtain the necessary follow-up, both due to socio -economically, as well as, due to specific ethnic beliefs and behaviors. I would suggest the authors consider the following manuscript that exemplifies the influence that being part of a particular community can influence screening participation and access to healthcare (Simion, L.; Rotaru, V.; Cirimbei, C.; Gales, L.; Stefan, D.-C.; Ionescu, S.-O.; Luca, D.; Doran, H.; Chitoran, E. Inequities in Screening and HPV Vaccination Programs and Their Impact on Cervical Cancer Statistics in Romania Diagnostics 2023 , 13, 2776. https://doi.org/10.3390/diagnostics13172776).

Having said that, I think this manuscript is of sufficient significance to the scientific community, it is well structured, with a clear methodology and discussion. However, I suggest a few changes:

I think the study has some limitations that were acknowledged by the authors (the most important being the older age of the Chinese respondents and the fact that the South Asians did not benefit from educational materials in their own language). The bias introduced by the age difference should be discussed in the results section, not in the discussion (results presented as such can be a bit misleading) – please correct this. The other issue, although discussed, may also introduce severe bias. Yes, it is true that English is an official language in many South Asian countries, but it is also true that it is usually spoken by more educated people. Since the study included people with a low level of education, this may be a consequence. Another limitation, in my opinion, would be that the proportion of Chinese respondents is lower than the proportion of the US Chinese population (the group is not truly representative of all Asian men living in the US).

Author response: Thank you very much for your helpful comments and critiques. We have moved the discussion of age differences to the results section. Additionally, we have omitted the statement that English is an official language in many South Asian countries due to the lack of data on the association between education level and English proficiency among South Asian Americans. Without such data, our original statement could be misleading, especially considering that, as the reviewer noted, education levels in our sample are generally low thereby not accurately represent the education levels of South Asians the US (Note: approximately 63% of South Asian men in our sample did not have at least a bachelor’s degree). We agree with your point and have noted in the Discussion section that our sample predominantly consisted of Vietnamese individuals, reflecting the demographic that Vietnamese is the largest Asian ethnic group in Houston, where our study was conducted. This composition does not align with the national proportions of Asian ethnic groups in the U.S., where Chinese individuals represent the largest ethnic group. We acknowledge this point as a limitation in the Discussion section.

In my opinion, a paragraph including some specific cultural peculiarities of the Asian population living in the US that may better explain the lack of tracking is warranted and should be added.

Author response: This is a valid point. However, the scarcity of research focused on Asian American men, particularly in identifying cultural characteristics distinct from those of other racial/ethnic groups, hampers our ability to pinpoint the specific cultural nuances within the Asian population in the US that may affect follow-up rates. Therefore, we have included a statement in the Conclusion section calling for research into the unique cultural characteristics of Asian American men in the context of cancer screening. Such studies could elucidate how these differences impact cancer screening behaviors.

One of the conclusions states: "Differences emerged in clinical abnormalities and baseline knowledge of prostate cancer." Please remove this as the first part can be explained by age difference rather than ethnic group and the second part needs to be further explained (Vietnamese group had a higher rate of correct answer although a level of education lower Please add a multivariate analysis of difference by educational attainment to the results section.

Author response: As requested, we have removed the sentence. Additionally, as requested, we have added a multivariate analysis of difference by educational attainment to the results section. Specifically, we found that both ethnicity and education level were significantly associated with pre-education knowledge scores (please see a table 3 that has been added to the Results section). Specifically, Chinese (p=.01) and South Asian (p<.001) individuals, as well as those without a bachelor's degree (p=.04), reported lower scores compared to their Vietnamese counterparts. However, the post-education analysis revealed that neither ethnicity (with p-values of .58 for Chinese and .35 for South Asians) nor educational level (p=.62) was significantly associated with post-education knowledge scores.

Table 3. Multivariate linear model assessing the association between ethnicity, education level, and pre- and post-education knowledge scores.

Knowledge at pre-education

Knowledge at post-education

Ethnic groupa

Coefficient

P-value

Coefficient

P-value

 Chinese

-1.17

.01

.24

.58

 South Asian

-2.98

<.001

-.72

.35

Educationb

1.01

.04

.23

.62

a reference group=Vietnamese. b 0=below a bachelor’s degree; 1=at least a bachelor’s degree.

Reviewer 2 Report

Comments and Suggestions for Authors

This is a well-written manuscript which provided a clear explanation of the research gaps, the overall project, and its importance to health care and public health professionals. My only recommendation would be to provide more up-to-date references - especially within the introduction. Many of the articles are more than 10 years old.

The main question addressed by the research is does education provided by the Prostate Outreach Project impact prostate cancer-related knowledge.

The topic addresses a gap in the literature related to prostate cancer knowledge among Asian men.

It provides context specific to Asian men.

The conclusions are consistent with the evidence and arguments presented and do they address the main question posed.

Updated references should be provided as many of the articles cited are more than 10 years old. 

Author Response

Reviewer 2

This is a well-written manuscript which provided a clear explanation of the research gaps, the overall project, and its importance to health care and public health professionals. My only recommendation would be to provide more up-to-date references - especially within the introduction. Many of the articles are more than 10 years old.

The main question addressed by the research is does education provided by the Prostate Outreach Project impact prostate cancer-related knowledge.

The topic addresses a gap in the literature related to prostate cancer knowledge among Asian men.

It provides context specific to Asian men.

The conclusions are consistent with the evidence and arguments presented and do they address the main question posed.

Updated references should be provided as many of the articles cited are more than 10 years old.

Author response: Thank you very much for your comments. We appreciate your concern regarding the age of the references we cited. Please find below two references published in 2022 that have been newly added to the manuscript. Please also note that the timeframe of our study, conducted between 2003 and 2008, was aligned with the American Cancer Society’s (ACS) cancer screening guidelines of 2002, which influenced the initial scope and direction of our research. We intentionally referenced materials from this period to contextualize our findings within the contemporary understanding and practices of prostate cancer screening. Notably, the USPSTF has indeed updated prostate cancer screening guidelines in 2008, 2012, and 2018. These updates are critical; however, our focus was to reflect the screening landscape as it was during our study period, to provide a historical baseline from which changes in screening practices could be assessed. Moreover, post-2018 USPSTF guidelines, which recommend a more individualized approach to prostate cancer screening, there has been a noticeable decline in the publication of epidemiological studies, which investigate adherence to screening recommendations, and intervention studies aimed at enhancing screening practices. Particularly challenging is our focus on Asian American men, a demographic that has been significantly underrepresented in cancer screening research. The scarcity of recent studies targeting the prostate health screening behaviors of this group further constrained our ability to include newer references. Most of the available literature on this topic dates back more than a decade, underscoring a gap in the research that our study aims to address. Our focus on this demographic highlights a critical gap in the literature, as most relevant studies predate the latest guideline changes. We trust that the reviewer will understand our perspective.

[3] Bergengren, O., et al., 2022 Update on Prostate Cancer Epidemiology and Risk Factors—A Systematic Review. European Urology, 2023. 84(2): p. 191-206.

[35] Van Poppel, H., et al., Serum PSA-based early detection of prostate cancer in Europe and globally: past, present and future. Nature Reviews Urology, 2022. 19(9): p. 562-572

Reviewer 3 Report

Comments and Suggestions for Authors

Dear authors,

a very relevant topic. 

I would like the group to discuss the following points:

i) if I understood everything right, the prostate cancer related knowledge was monitored before and IMMEDIATELY after watching the video. If you measure learning performance directly after an instructional video, everyone would expect it to be high. Isn't it much more interesting to measure learning performance at intervals of several weeks? Could this not possibly explain the low rate of 15.2% of people who did not attend the medical follow-up?

ii) In general, screening programs for prostate cancer will gain in importance worldwide. However, I consider the combination of PSA and DRE to be critical and outdated. We are all aware of the lack of specificity of the PSA test, which can lead to overtreatment. In addition, you will also miss relevant carcinomas with a threshold of 4 ng/ml. The DRE can only detect advanced carcinomas. I would ask you to research the literature on current worldwide screening programs, including their cost-effectiveness and give reasons why PSA and DRE were chosen

Author Response

i) if I understood everything right, the prostate cancer related knowledge was monitored before and IMMEDIATELY after watching the video. If you measure learning performance directly after an instructional video, everyone would expect it to be high. Isn't it much more interesting to measure learning performance at intervals of several weeks? Could this not possibly explain the low rate of 15.2% of people who did not attend the medical follow-up?

Author response: Thank you for your insightful feedback. Indeed, the assessment of post-education prostate cancer-related knowledge was conducted immediately after participants viewed the educational video. This methodological decision was driven by the practical constraints of conducting the study within community settings, where it was not feasible to follow up with participants who were recruited on-site for subsequent assessments at multiple times. We acknowledge this limitation and have elaborated in the Discussion section that the immediate assessment might not fully reflect the participants' long-term knowledge retention, which could partly explain the low follow-up rates observed in our study.

ii) In general, screening programs for prostate cancer will gain in importance worldwide. However, I consider the combination of PSA and DRE to be critical and outdated. We are all aware of the lack of specificity of the PSA test, which can lead to overtreatment. In addition, you will also miss relevant carcinomas with a threshold of 4 ng/ml. The DRE can only detect advanced carcinomas. I would ask you to research the literature on current worldwide screening programs, including their cost-effectiveness and give reasons why PSA and DRE were chosen.

Author response: Thank you for bringing up this important point. Please note that our study was conducted between 2003 and 2008, aligning with the American Cancer Society’s (ACS) cancer screening guidelines of 2002. During that period, the ACS recommended annual PSA and/or DRE screenings for men aged 50 and older, suggesting that individuals at high risk might need to start screening at age 40. We have acknowledged the reliance on these now-outdated guidelines as a limitation in the Discussion section of our manuscript.

Round 2

Reviewer 1 Report

Comments and Suggestions for Authors

The authors made in my opinion sufficient relevant changes fo r the article to be published in the current form. They have responded completely to all my concerns.